# Inductive Determination of Rate-Reaction Equation Parameters for Dislocation Structure Formation Using Artificial Neural Network

**DOI:** 10.3390/ma16052108

**Published:** 2023-03-05

**Authors:** Yoshitaka Umeno, Emi Kawai, Atsushi Kubo, Hiroyuki Shima, Takashi Sumigawa

**Affiliations:** 1Institute of Industrial Science, The University of Tokyo, 4-6-1 Komaba, Meguro-ku, Tokyo 153-8505, Japan; 2Department of Environmental Sciences, University of Yamanashi, 4-4-37, Takeda, Kofu, Yamanashi 400-8510, Japan; 3Department of Energy Conversion Science, Graduate School of Energy Science, Kyoto University, Sakyo-ku, Kyoto 606-8501, Japan

**Keywords:** reaction–diffusion model, dislocation structure, fatigue, artificial neural network, multiscale simulation, machine learning

## Abstract

The reaction–diffusion equation approach, which solves differential equations of the development of density distributions of mobile and immobile dislocations under mutual interactions, is a method widely used to model the dislocation structure formation. A challenge in the approach is the difficulty in the determination of appropriate parameters in the governing equations because deductive (bottom-up) determination for such a phenomenological model is problematic. To circumvent this problem, we propose an inductive approach utilizing the machine-learning method to search a parameter set that produces simulation results consistent with experiments. Using a thin film model, we performed numerical simulations based on the reaction–diffusion equations for various sets of input parameters to obtain dislocation patterns. The resulting patterns are represented by the following two parameters; the number of dislocation walls ($p_{2}$), and the average width of the walls ($p_{3}$). Then, we constructed an artificial neural network (ANN) model to map between the input parameters and the output dislocation patterns. The constructed ANN model was found to be able to predict dislocation patterns; i.e., average errors in $p_{2}$ and $p_{3}$ for test data having 10% deviation from the training data were within 7% of the average magnitude of $p_{2}$ and $p_{3}$. The proposed scheme enables us to find appropriate constitutive laws that lead to reasonable simulation results, once realistic observations of the phenomenon in question are provided. This approach provides a new scheme to bridge models for different length scales in the hierarchical multiscale simulation framework.

## 1. Introduction

Fatigue is a common fracture mode in metal and accounts for a substantial fraction of failure cases in real industrial products. It is therefore demanded to fully understand the mechanism of fatigue fracture. In particular, there is still much room for investigations of the mechanism of fatigue crack formation under cyclic loading. It is widely understood that the fatigue crack formation in macroscopic metal materials originates in the persistent slip band (PSB) formed as a result of self-organization of dislocation structures [1]. Nevertheless, the PSB formation mechanisms proposed thus far have room for further examination and assessment, urging investigation by modeling and simulation. Moreover, recent experimental studies of fatigue in nanometer- or submicron-sized materials by Sumigawa et al. [2,3] indicate the possibility of unveiled mechanisms of fatigue at the nanometer and submicron scales, or “nano–micro fatigue”. As fatigue fracture was observed in a specimen smaller than the dimension of PSB, which suggests that fatigue fracture can occur without the presence of PSB, the variation of self-organized dislocation patterns due to size effect should play a key role in nano–micro fatigue [2,4]. This finding also urges modeling and simulation to reveal unknown complicated mechanisms lying behind the dislocation structure formation.

Walgraef and Aifantis proposed a phenomenological model of the dislocation pattern formation in metal under cyclic loading based on the rate-reaction (reaction–diffusion) theory [5,6,7,8,9,10,11,12,13,14]. The reaction–diffusion equation approach, which solves differential equations of the development of density distributions of mobile and immobile dislocations under mutual interactions, has been widely used to simulate the dislocation structure formation and was found to be useful to discuss its mechanisms. A challenge in such a phenomenological model is, however, the difficulty in the determination of appropriate parameters in the governing equations. Parameters are often fitted so that the simulation results become consistent with experimental observations of the phenomenon in question, but this can be daunting when the governing equations contain a number of parameters.

An alternative way would be bottom-up (deductive) determination based on a different physical model covering a smaller length scale. For example, dislocation mobility used in the discrete dislocation dynamics can be obtained by a molecular dynamics simulation of a single dislocation. Although this scheme may look straightforward, consistency with experimental facts is not guaranteed because the simulation at the lower scale may produce nontrivial deviation from reality due to technical constraints such as the limitation of spatial and temporal scales of the simulation setup. Moreover, the deductive approach must rely on a multi-story hierarchy of multiscale models if the phenomenological model is on the macroscopic side, which can make the parameter determination substantially prone to accumulated deviation.

In this study, we propose an inductive approach for the parameter determination utilizing machine-learning. Using a thin film model, we performed numerical simulations of dislocation structure formation based on the Walgraef–Aifantis (WA) model using various sets of input parameters. The results of dislocation structures (density distributions) were characterized with devised algorithms. Then, we constructed an artificial neural network (ANN) model that predicts the output dislocation patterns from the input parameters. The application of the proposed scheme for finding appropriate constitutive laws consistent with experiments was discussed. This new scheme paves the way for bridging models for different length scales in the hierarchical multiscale simulation framework.

## 2. Methodology

### 2.1. Reaction-Diffusion Equation by Walgraef-Aifantis

Walgraef and Aifantis proposed a reaction–diffusion model (which we call the WA model hereafter in this paper) to describe temporal change in dislocation densities, which has a long history and is widely used for its convenience. In this model, dislocations are divided into two categories; i.e., the mobile and immobile dislocations. The former is free to move as a response to stress exerted in the slip plane, while the latter is trapped or moves slowly. The mobile and immobile dislocation densities in space x and time t are, respectively, described by $\rho_{m}\left(x,t\right)$ and $\rho_{i}\left(x,t\right)$. Temporal evolution of the dislocation density functions is obtained by solving the following parallel non-linear partial differential equations [14]:(1)$$\frac{\partial \rho_{i}}{\partial t}=D_{i}\frac{\partial^{2}\rho_{i}}{\partial x^{2}}+\alpha \left(\rho_{0i}-\rho_{i}\right)-\beta \rho_{i}+\gamma \rho_{m}\rho_{i}^{2}$$
(2)$$\frac{\partial \rho_{m}}{\partial t}=D_{m}\frac{\partial^{2}\rho_{m}}{\partial x^{2}}+\beta \rho_{i}-\gamma \rho_{m}\rho_{i}^{2}$$

The first terms of the right-hand side of the two equations represent diffusion-like behavior of the mobile and immobile dislocations, with $D_{i}$ and $D_{m}$ being constants for the strength of diffusivity. Pinning-up of newly produced dislocations during the formation of PSBs is expressed by $\alpha \left(\rho_{0i}-\rho_{i}\right)$, where α represents the annihilation rate and $\rho_{0i}$ is a constant value describing the source of immobile dislocations, which is assumed to exist uniformly in the system. Release of immobile dislocations from the dislocation forest is described by $\beta \rho_{i}$, where β designates the dislocation release from the forest. Capture of mobile dislocations by immobile dipoles is represented by the nonlinear term of $\gamma \rho_{m}\rho_{i}^{2}$, with γ being the capture rate.

While the two parallel equations of the evolution of the dislocation density functions include five parameters (WA parameters; $D_{i},\;D_{m},\;\alpha ,\;\beta ,\;\gamma$), not all of them are independent, meaning that the five parameters should not be arbitrarily determined. According to Schiller et al. [15], the following relations hold among the pinning-up rate, the release rate and the diffusivity strengths:(3)$$\alpha =\frac{D_{i}}{l_{i}^{2}}$$
(4)$$\gamma =\frac{v_{m}^{2}}{2\rho_{0i}^{2}D_{m}}$$
where $l_{i}$ and $v_{m}$ are the mean free path of immobile dislocations, and the effective velocity of mobile dislocations considering trapping by obstacles, respectively.

In addition, it is known that there are two bifurcations in the dislocation pattern depending on the value of β. If β equals or exceeds a critical value of $\beta_{H}$, the dislocation pattern oscillates with time, which is called the Hopf bifurcation. The other critical value is $\beta_{c}$ (assuming $\beta_{c}<\beta_{H}$), at which the Turing instability occurs; i.e., the dislocation pattern is formed if β exceeds $\beta_{c}$. According to stability analysis, $\beta_{H}$ and $\beta_{c}$ are in relation with some parameters as [14]
(5)$$\beta_{c}={\left(\sqrt{\alpha }+\sqrt{\frac{cD_{i}}{D_{m}}}\right)}^{2}$$
(6)$$\beta_{H}=\alpha +c$$
(7)$$c=\gamma \rho_{0i}^{2}$$

The five WA parameters with the consideration of the abovementioned traits were changed as follows:

$D_{i}$ was set to be $10^{-4},10^{-3.5}\;\mathrm{or}\;10^{-3}\;$µm^2^/s.

$D_{m}$ was set such that $D_{i}/D_{m}=0.2\times 10^{-2},\;0.5\times 10^{-2}\;\mathrm{or}\;10^{-2}$ because $D_{i}/D_{m}$ should be at least 10^−2^ [15].

β was set to be $\beta_{1},\beta_{2}\;\mathrm{or}\;\beta_{3}$, where $\beta_{1}=0.9\;\beta_{c},\;\beta_{2}=\beta_{c}+\frac{\beta_{H}-\beta_{c}}{3}\;\mathrm{and}\;\beta_{3}=\beta_{2}+\frac{\beta_{H}-\beta_{c}}{3}$ (NB: $\beta_{1}<\beta_{c}<\beta_{2}<\beta_{3}<\beta_{H}$).

α and γ are determined according to Equations (3) and (4), with $l_{i}=10^{-2}\;\mu m$,$\;v_{m}=10\;\mu m/s$ and $\rho_{0i}=0.5\;{\mu m}^{-2}$ [15], which is an average value of the initial distribution of *ρ*_i_.

Therefore, 27 parameter sets were used in total.

As described above, $\rho_{i}\;\mathrm{and}\;\rho_{m}$ are functions of space coordinate x and time t, meaning that we consider a one-dimensional distribution of dislocations. x is in the range 0≤x≤l, where l indicates the thickness of the space. In our simulation, we set l=1.0 μm. At the boundaries of the space, the spatial derivatives of the dislocation densities are assumed to be zero; i.e., $\frac{\partial \rho_{i,m}}{\partial x}=0$ at x=0, l.

Here, we explain two schemes to construct initial dislocation distributions from which the parallel reaction–diffusion calculations start. One is a simple way to use random fractional values of f(0<f<1) as the initial dislocation density (Scheme A). The other is a devised scheme to give more smooth but random distributions (Scheme B). There, the initial dislocation distributions were generated by superposing sinusoidal waves with various (given) wavenumbers and random amplitudes, controlled to satisfy the given boundary condition and minimum/maximum values. The detailed procedure of Scheme B is explained in Appendix A. From these initial distributions, we solved the parallel partial differential equations (Equations (1) and (2)) numerically using the Euler method with a time step of $1.0\times 10^{-6}\;s$.

### 2.2. Characterization of Resulting Dislocation Structure

After a sufficient number of iterations of numerical integral (time development) of Equations (1) and (2), we obtain converged dislocation densities ($\rho_{i}\;\mathrm{and}\;\rho_{m})$. Since the density distribution of immobile dislocations should represent the dislocation pattern formed as a result of diffusion and reaction of mobile and immobile dislocations, we analyze the form of $\rho_{i}$. Figure 1 schematically shows two typical distribution patterns of immobile dislocations. Figure 1a depicts the case where we find peaks of dislocation density aligned with low-density areas lying in between, which can be regarded as the formation of the wall structure. Thus, the peaks in $\rho_{i}$ will be called “walls” hereafter in this paper. In contrast, no characteristic shape is found in some cases such as Figure 1b, indicating no formation of self-organized dislocation patterns.

Now, we need an algorithm to extract features of the distribution pattern from the function $\rho_{i}\left(x\right)\;\left(0\leq x\leq l\right)$, namely; (a) the presence of a self-organized pattern; (b) the number of walls; and (c) the average width of walls. Our algorithm works as follows: It is regarded that a self-organized pattern is formed when more than 50% of $\rho_{i}\left(x\right)$ exceeds the threshold $\rho_{\mathrm{th}}≔\rho_{\min}+0.05\;\left(\rho_{\max}-\rho_{\min}\right)$, where $\rho_{\max}$ and $\rho_{\min}$ are the maximum and minimum values of $\rho_{i}\left(x\right)$. By detecting points where the curve $y=\rho_{i}\left(x\right)$ and $y=\rho_{\mathrm{th}}$ intersect each other, the number of walls can be counted. The width of a wall is defined as the width of a continuous region where $\rho_{i}\left(x\right)\geq \rho_{\mathrm{th}}$.

### 2.3. Mapping of WA Parameters and Resulting Dislocation Structure

#### 2.3.1. Structure of Artificial Neural Network model

Our ANN model consists of five layers including the input and output layers. The number of nodes on each layer is 5 → 6 → 6 → 4 → 3. The input layer has five nodes corresponding to WA parameters ($D_{i},\;D_{m},\;\alpha ,\;\beta \;\mathrm{and}\;\gamma$), respectively. The output layer has three nodes, which give quantities ($p_{1},p_{2}\;\mathrm{and}\;p_{3}$) for characterization of the resulting dislocation pattern. $p_{1}$ is a Boolean value representing whether a dislocation wall structure is formed. $p_{2}\;\mathrm{and}\;p_{3}$ are the number of walls [${\mu m}^{-1}$] and the average width [nm] of the formed walls, respectively.

Figure 2 shows a schematic illustration of the ANN architecture. The present ANN model is based on a typical feed-forward network, consisting of five layers; one input layer (hereafter, referred to as Layer 0), three internal layers (Layers 1, 2, and 3), and one output layer (Layer 4). Each layer consists of nodes. The numbers of nodes in Layer n, $N_{n}$, are set to 5, 6, 6, 4, and 3 for Layers 0, 1, 2, 3, and 4, respectively. The nodes in Layer 0 (input) and Layer 4 (output) are corresponding to the WA parameters (*D*_i_, *D*_m_, *α*, *β*, *γ*) and the characterization parameters ($p_{1},p_{2},p_{3}$), respectively. The state of each node is given by a real number, and hereafter we refer to the state of the *q*-th node in the n-th layer as $x_{q}^{n}$. The states of nodes in the input layer (n=0), $x_{q}^{0}\left(q=1,\;2,\;3,\;4,\;5\right)$, are given by the common logarithms of the WA parameters:(8)$$\begin{array}{l} x_{1}^{0}={log}_{10}D_{i},\\ x_{2}^{0}={log}_{10}D_{m},\\ x_{3}^{0}={log}_{10}\alpha ,\\ x_{4}^{0}={log}_{10}\beta ,\\ x_{5}^{0}={log}_{10}\gamma . \end{array}$$

Note that we adopted a logarithm of the parameters instead of the parameters themselves because the parameters are expected to vary in a wide range (by several digits). The states of the internal and output layers are determined by the previous layer as [16]
(9)$$x_{q}^{n}=f_{n}\left(w_{q0}^{n}+\overset{N_{n-1}}{\underset{r=1}{\sum }}w_{qr}^{n}x_{r}^{n-1}\right),$$
where $w_{qr}^{n}$ denotes the weight parameter, and $w_{qr}^{n}$ is the bias parameter ($n=1–4;q=1,\ldots ,N_{n};r=1,\ldots ,N_{n-1}$), which are the internal parameters to be optimized by machine learning. Therefore, the total number of the parameters is 121. The function $f_{n}\left(x\right)$ represents the activation functions defined as [16]
(10)$$f_{n}\left(x\right)=\{\begin{array}{cc} \frac{1}{1+e^{-x}} & \left(n=1,\;2,\;3\right)\\ x & \left(n=4\right) \end{array}.$$

The node states of the output layer are interpreted as the predicted characterization parameters, i.e., $p_{q}=x_{q}^{4}\left(q=1,2,3\right)$. Note that the Boolean parameter $p_{1}$ deals with a real number in the ANN model for simplicity (If a wall structure is formed, then $p_{1}=1$; otherwise $p_{1}=0$). In this description, the value of $p_{1}$ can be interpreted as the probability of formation of a wall structure.

The numbers of intermediate layers and nodes on the layer are arbitrarily chosen, but should affect the performance of the ANN model. This will be discussed later in this paper.

#### 2.3.2. Training of ANN

The reaction–diffusion equations were solved for the parameter sets and the initial structures described in Section 2.1 (i.e., 27×10=270 cases for each scheme). Among these cases, we found that the final *ρ*_m_ had negative values for the cases with $D_{i}/D_{m}=0.2\times 10^{-2}$ and $D_{i}=10^{-3}\;{\mu m}^{2}/s$. This was presumably because $D_{m}$ was relatively large, resulting in numerical errors in solving the partial differential equations with the Euler method. Excluding these parameter sets, we used 240 (=24×10) cases as training data of the ANN model for each scheme. Then, the ANN model was trained to map the input WA parameters to the resulting dislocation structure ($\;p_{1},p_{2}\mathrm{and}\;p_{3}$).

The loss function, *L*, which represents deviation of the ANN prediction from actual results, is set to be
(11)$$L=\overset{n}{\underset{k=1}{\sum }}\overset{3}{\underset{i=1}{\sum }}{\left(p_{i,k}-p_{i,k}^{0}\right)}^{2}$$
where $p_{i,k}$ indicates $p_{i}$ of the case k out of the combinations of parameter sets and initial distributions by either Scheme A or B. n=240 (24 parameter sets and 10 initial distributions) is the total number of the cases. The WA model parameters were optimized to reduce the loss function with the steepest descent method.

#### 2.3.3. Test of ANN

To examine the predictability of the trained ANN, i.e., the reliability of the prediction when a parameter set deviates from training datasets, we prepared test datasets (denoted with $\hat{\;}$) using a predetermined value indicating the amount of deviation, $\Delta_{i,m}$, as follows: First, ${\hat{D}}_{i}$ and ${\hat{D}}_{m}$ were determined as
(12)$${\hat{D}}_{i,m}=D_{i,m}+R_{i,m}\Delta_{i,m}$$
with $R_{i}$ and $R_{m}$ randomly taking integer values −1, 0 or 1. Here, all combinations of $R_{i}$ and $R_{m}$ excluding $R_{i}=R_{m}=0$ (i.e., eight cases) were produced with the same probability. Next, using the determined ${\hat{D}}_{i}$ and ${\hat{D}}_{m}$ we obtained $\hat{\alpha }$ and $\hat{\gamma }$ as follows:(13)$$\hat{\alpha }=\frac{{\hat{D}}_{i}}{l_{i}^{2}}.\;$$
(14)$$\hat{\gamma }=\frac{v_{m}^{2}}{2\rho_{0i}^{2}{\hat{D}}_{m}}$$
where the values of *l*_i_, *v*_m_ and *ρ*_0i_ are the same values shown in Section 2.1. Finally, $\hat{\beta }$ was determined in the following way so that the magnitude correlation among $\beta ,\;\beta_{c}\;\mathrm{and}\;\beta_{H}$ was kept: After calculating ${\hat{\beta }}_{c}$ and ${\hat{\beta }}_{H}$ based on ${\hat{D}}_{i}$ and ${\hat{D}}_{m}$, $\hat{\beta }$ was obtained as
(15)$$\{\begin{array}{cc} \hat{\beta }=0.9{\hat{\beta }}_{c}+R^{\prime }\Delta & :\beta <\beta_{c}\\ \hat{\beta }={\hat{\beta }}_{c}+\frac{\beta -\beta_{c}}{\beta_{H}-\beta_{c}}\left({\hat{\beta }}_{H}-{\hat{\beta }}_{c}\right)+R^{\prime }\Delta & :\beta_{c}<\beta <\beta_{H} \end{array}$$
where $R^{\prime }$ randomly takes integer values −1, 0 or 1 and Δ represents the amount of deviation.

To examine the predictability of the ANN model according to the deviation magnitude of test datasets from the training datasets, we set $\Delta_{i,m}\;\mathrm{and}\;\Delta$ to be 0.1, 1 or 10 % of the corresponding parameter values.

## 3. Results and Discussion

### 3.1. Training of ANN

Figure 3 shows changes in the loss function during the steepest descent iterations. In both the cases with initial structures by Scheme A and Scheme B, the loss function was successfully reduced. This means that the training of the ANN model on the prepared datasets was achieved.

Figure 4 compares dislocation structures ($p_{1},p_{2}\;\mathrm{and}\;p_{3}$) predicted by the trained ANN and the actual WA results. Note that the predicted $p_{1}$ can take fractional (non-integer) values, which makes the deviation of the predicted values from the actual $p_{1}$ a little conspicuous (up to 0.28 and 0.29 for Schemes A and B, respectively). It is however only a few points that show a relatively large deviation. It is noticed therefore that the predicted and actual values are overall in good agreement.

### 3.2. Evaluation of ANN with Test Datasets

Figure 5 compares predicted dislocation structures ($p_{1},p_{2}\;\mathrm{and}\;p_{3}$) with the ANN and actual WA solutions for the deviated parameter sets. To quantitatively assess the validity of the ANN, errors in the ANN prediction of $p_{2}\;\mathrm{and}\;p_{3}$ from the actual WA solutions were calculated and compared with the magnitude of $p_{2}\;\mathrm{and}\;p_{3}$, respectively. The average errors for the parameter sets with 10% deviation (test data) were found to be within 7% of the average magnitude of $p_{2}\;\mathrm{and}\;p_{3}$. The comparison overall demonstrates a good performance of the ANN model giving predictions in a good agreement with the actual WA results, with exceptions at relatively large average wall widths ($p_{3}$). These deviations are, however, reasonable because the points showing the large deviation are from the cases that were excluded from the training of the ANN (i.e., $D_{i}/D_{m}=0.2\times 10^{-2}$ and $D_{i}=10^{-3}\;{\mu m}^{2}/s$). These cases have the largest value of $D_{m}=0.5\;{\mu m}^{2}/s$ among the test datasets, which was presumably a reason behind the large deviation because machine learning is basically not suitable for extrapolation.

Although the trained ANN does not seem to work well for extrapolation as shown above, its predictability for 10% deviation from the training datasets demonstrates the good performance of the constructed ANN. It is presumably possible to make the ANN model more robust and reliable to cover a wider area in the parameter space by providing more training datasets. It is however not the objective of this study to construct such a robust ANN model to bypass the WA diffusion–reaction equation calculation. We rather aim to suggest the possibility of mapping between the input parameters of a simulation model and its results using machine learning.

### 3.3. Possibility of Inductive Construction of a Simulation Model

The successful demonstration of the input–output mapping paves the way for the inductive determination of input parameters of a phenomenological simulation model. Once the mapping that links between the input parameters and the results is achieved, it is possible to select a parameter set that produces a desired simulation result. Now, if we have experimental results of a specific phenomenon and a reliable simulation model (e.g., governing equations to model the phenomenon), we can conjecture what the simulation result should look like. Then, we can pick a parameter set that gives a simulation result consistent with the experimental observation. In other words, this way is to find parameters in the simulation model as a reverse problem. This inductive scheme, which can be regarded as a top-down determination of parameters, provides an alternative measure to determine constitutive equation parameters, which is often challenging to carry out in a bottom-up (deductive) manner.

The proposed scheme of inductive determination of simulation models can be regarded as one example of a physics-informed neural network (PINN) [17,18,19,20,21]. Raissi et al. proposed the following three types of PINN:(A)Finding solution of partial differential equationsThe function forms and parameters of partial differential equations (PDEs) are known. Initial and boundary conditions are given at discrete sampling points. A neural network mimicking the solution of the PDEs is to be found.(B)Finding parameter of partial differential equationsThe function forms of PDEs are known while their parameters are unknown. The solution of the PDEs is given at discrete sampling points. A neural network as the solution of the PDEs is to be found, resulting in the determination of the PDE parameters.(C)Finding latent physical quantities in observationsThe function forms and parameters of PDEs are known. A physical quantity in the considered system is given by observation. A neural network giving the observed physical quantity and other (latent) physical quantities appearing in the PDEs is to be found so that the prediction of the quantities is consistent with the observation and the PDEs.

Among these PINN types, our approach may fall in (B) where parameters in PDEs are found by means of a neural network model.

It should be noted here, however, that what was represented by the ANN in our approach is not the mapping between the parameters and the PDE solution, but that between the parameters and the values quantifying the PDE solution. In other words, we presented an original scheme to quantify the dislocation density distribution as the WA solution while we adopted a simple ANN model that maps between the input and output sets of scalar values.

### 3.4. Integration of Deduction and Induction Approaches in Multiscale Modeling

The scheme of inductive determination of simulation parameters should not be limited to the WA model as demonstrated in this study, but can be applied to any other simulation models in general. A simulation model usually consists of governing equations that have some parameters, and its solution can be obtained once the parameters and the initial conditions are given. Thus, it is possible to apply the proposed scheme to any simulation model and obtain mapping between the parameters and the solution, making the inductive determination of the parameters possible once the desired (i.e., consistent with observed facts) simulation results are known.

This approach may be extended to realize a reasonable link between simulation models covering adjoining length scales as schematically shown in Figure 6. Let us take a situation, for example, where we deal with a material behavior that requires two scale models, the lower of which can be treated with the atomistic model (e.g., molecular dynamics) and the larger is described by a phenomenological model (e.g., phase field). Molecular dynamics can simulate material behaviors using interatomic potentials non-empirically constructed with first-principles calculations to obtain characteristic material properties such as diffusion coefficients, dislocation mobility, critical stress for crystal slips, etc., which is a so-called a bottom-up (deductive) evaluation of material properties.

This deductive/bottom-up scheme is often applied for scale-bridging in hierarchical multiscale simulation models [22]. For example, material properties at the nanometer scale obtained with atomistic model calculations can be put into an upper-scale (mesoscopic) model such as dislocation dynamics to conduct a larger scale (but coarser resolution) simulation. Similarly, the mesoscopic model can evaluate material properties at the corresponding scale, which may be fed to a macroscopic model. This way one can build a hierarchical multiscale model where different length scales are interconnected by the bottom-up fashion. The problem is, however, that the evaluation in the material properties may contain some errors, and the errors can accumulate if the bottom-up scale-bridging is repeated, leading to substantial deviation from reality at the macroscopic scale.

Accepting that the properties evaluated by a simulation model inevitably contain some amount of errors, we may adjust the obtained properties so that the models on the different stories of the multiscale hierarchy are mutually interconnected and consistent with experimental observations, which are usually given at the largest end of the hierarchy. The inductive determination of model parameters can be utilized for such objectives. The inductive scheme makes it possible to find possible parameter sets that produce results consistent with the experiment, i.e., the top-down determination of reliable parameters. When this is combined with the bottom-up evaluation of material properties, one can find constitutive laws of the material that are consistent with the experiment and also based on physics. The deduction–induction integration may be a promising new concept for the hierarchical multiscale simulation because it can eliminate common problems in the top-down (i.e., constitutive laws are not physics-based) and bottom-up (i.e., results can be deviated substantially from experiment) scale-bridging.

## 4. Conclusions

An inductive approach for the determination of appropriate parameters in simulation models by means of machine learning is presented and demonstrated for a rate-reaction model of dislocation structure formation. Using the reaction–diffusion equations of mobile and immobile dislocation density distributions proposed by Walgraef and Aifantis, we calculated dislocation wall formation in a one-dimensional model with various predetermined parameter sets.

After obtaining resulting dislocation wall structures with extensive different parameter sets, an ANN model was constructed to reproduce the characterization of the dislocation wall structures as a function of the parameter set. The constructed ANN presented a good predictability with test datasets; i.e., average errors for the parameter sets with 10% deviation from training data were found to be within 7% of the average magnitude of the target data, although non-negligible deviation was found for extrapolation. The presented scheme of inductive determination of simulation model parameters can be regarded as a top-down approach to find an appropriate constitutive laws of materials. This approach provides a new scheme to bridge models for different length scales in the hierarchical multiscale simulation framework.

## Figures and Tables

**Figure 1 materials-16-02108-f001:**
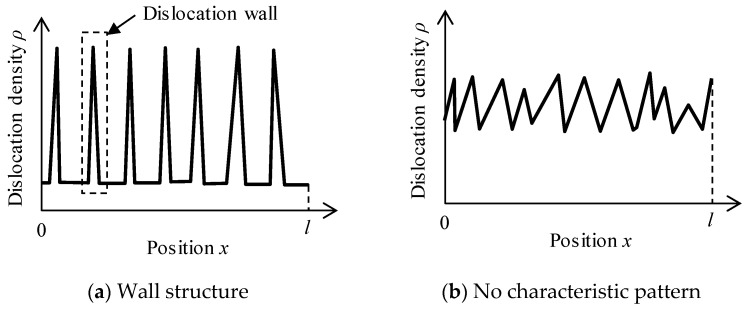
Schematics of typical dislocation density distributions ($\rho_{i}$). (**a**): Aligned peaks with low-density areas in between, indicating formation of wall structure. (**b**): No self-organized dislocation pattern.

**Figure 2 materials-16-02108-f002:**
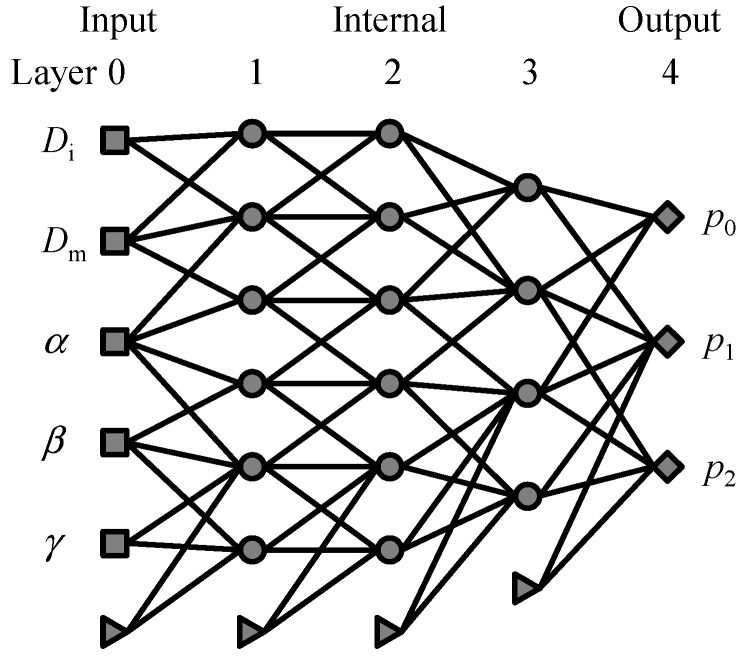
Schematic of ANN architecture. Nodes in each layer are indicated as square, circle, diamond, and triangle for the input layer, internal layers, output layer, and bias, respectively. Not all connections between adjacent layers are drawn.

**Figure 3 materials-16-02108-f003:**
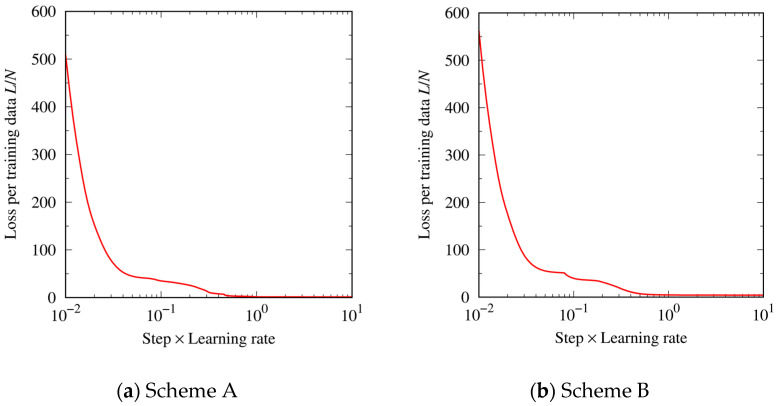
Loss function in (**a**) Scheme A and (**b**) Scheme B. The horizontal axis shows the value obtained by Step (max step is 10^8^) × Learning rate (10^−7^). The vertical axis shows the loss per training data of the ANN model (*N* = 240).

**Figure 4 materials-16-02108-f004:**
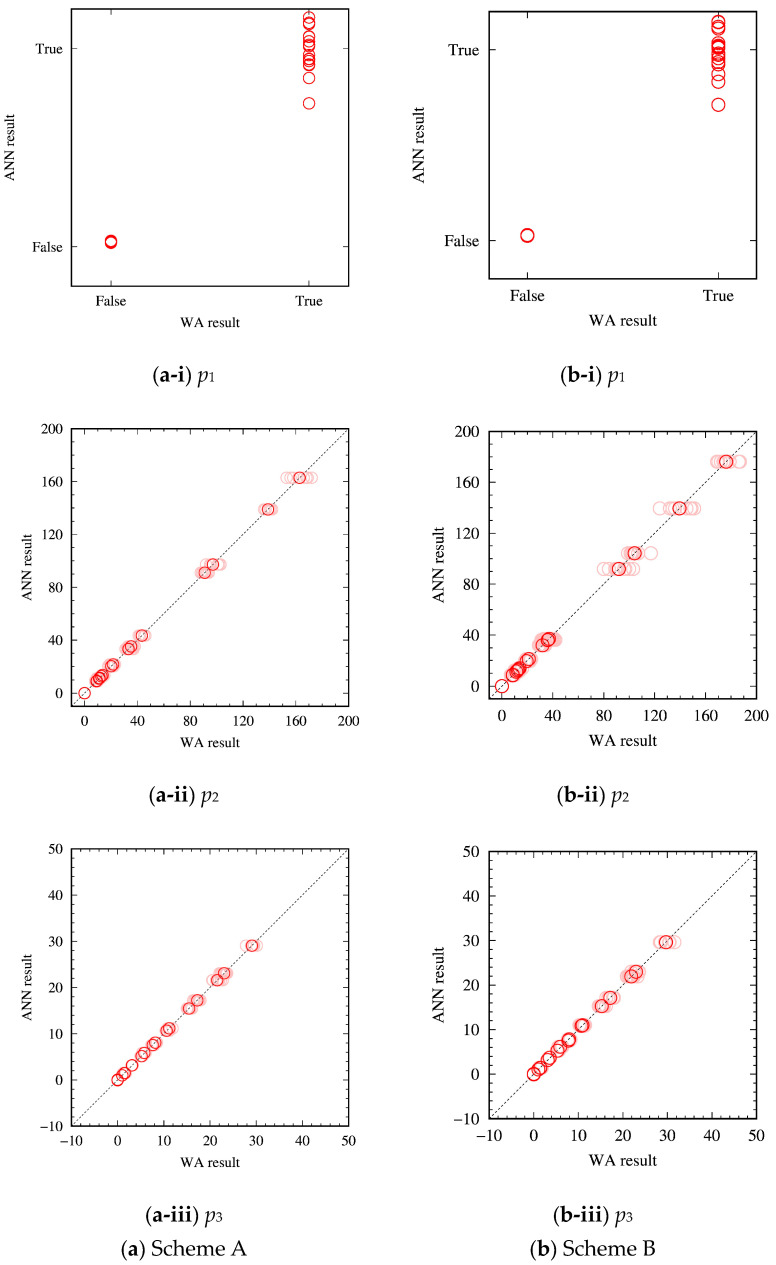
Comparison of predicted dislocation structures with the ANN and WA solutions for the training parameter sets of (**a**) Scheme A and (**b**) Scheme B. (i), (ii) and (iii) show Boolean values representing whether a dislocation wall structure is formed, *p*_1_, the number of walls, *p*_2_, and the average width of the formed walls, *p*_3_, respectively. In (ii) and (iii), the red and faded red plots show the results after and before averaging with the same initial structure, respectively.

**Figure 5 materials-16-02108-f005:**
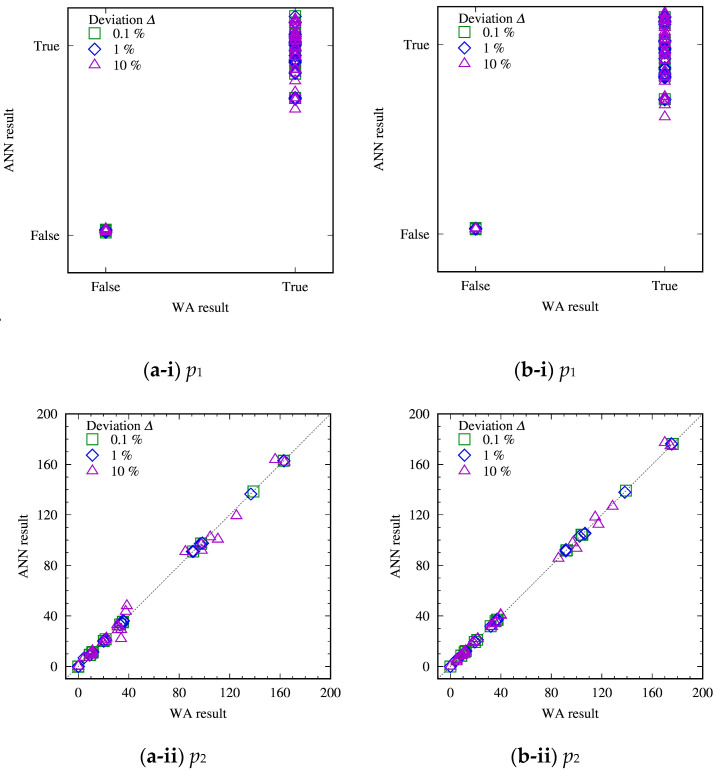
Comparison of predicted dislocation structures with the ANN and WA solutions for the deviated parameter sets of (**a**) Scheme A and (**b**) Scheme B. (i), (ii) and (iii) show Boolean values representing whether a dislocation wall structure is formed, *p*_1_, the number of walls, *p*_2_, and the average width of the formed walls, *p*_3_, respectively. The square, diamond and triangle plots show the amount of deviation *Δ* = 0.1, 1, and 10%, respectively.

**Figure 6 materials-16-02108-f006:**
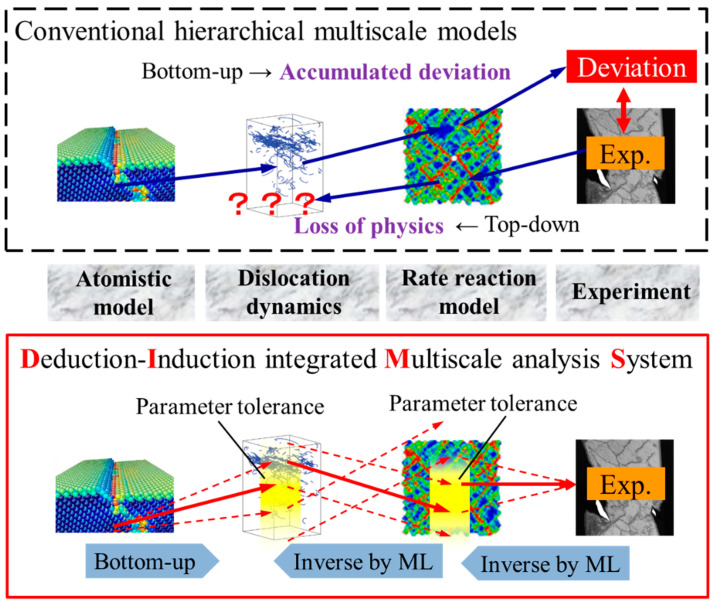
Schematic explaining the concept of suggested integration of deduction–induction approaches compared with conventional hierarchical multiscale models.

## Data Availability

The data presented in this study are available on request from the corresponding author.

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
