# Peer review of "Inductive Determination of Rate-Reaction Equation Parameters for Dislocation Structure Formation Using Artificial Neural Network"

_materials, 2023, doi:10.3390/ma16052108_

Round 1

Reviewer 1 Report

From my perspective, the topic is very interesting, but the overall quality should be improved:

1.    I recommend adapting the title to the scope of the work.

2.    Abstract - provide some numerical values as properties. It looks general.

3.    The introduction part includes a short story until it comes to the main objective of this paper. I think it should be written with more details to make it clear and exact.

4.    The Reviewer recommends the Authors to rewrite the last paragraph of the introduction section in a way to stress the novelty.

5.    Please replace the active tense, starting with “WE”, with the passive tense, more academic!

6.    Please pay attention to the quotation standards for references.

7.    The criteria of proposed model selection and author's assumption are not explained.

8.    Some equations in this paper that were not proposed by the author and were written without any references. Please check this point.

9.    Training, testing and validating process of ANN model should be explained in terms of used data. In addition, the flow chart of the proposed ANN is important.

10. The results  part of the paper is good; however, authors are advised to draw comparisons with previous literature to justify the results.

11. No need to discuss your results in the conclusions. The conclusion of the study needs to be rewritten. The conclusion should be quantified.

12. A few references need to be updated with some recent papers published in the last years. What were your entry and exit criteria for selecting articles? Did you check all the articles, books, theses? Did you check the articles written in any language? In what year did you search (range)?

13. The manuscript is overall well-written. However, there are many typesetting and grammatical errors in the text that should be corrected.

Reviewer 2 Report

There is no novelty in that paper. In the first part of the paper, from Section 1 to Section 3.2, the generation of a synthetic data set and machine learning are totally uncoupled. The proposed artificial neural network (ANN) is quite basic and its loss too. This ANN do not account for the time dependancy of the physics-based predictions. There is no discussion about the learning bias related to the restriction of the WA model to a 1D model. The authors claim that the proposed approach "paves the way for bridging models of different length scales", but this is not supported by any equations neither by numerical results. Section 3.3 and section 3.4 are not supported by new theoretical results neither numerical results in this paper.

Reviewer 3 Report

1. The title should be shortened.

2. A nomenclature should be added.

3. Fig4, it mentions that there are a few points with large deviations. Could you show exactly how much the deviation is?

4. Figure 6 provides a good illustration, but there is no comparison actually provided. Could you simulate in both conventional and proposed ways and provide a comparison?

5. Line 397 says experimental observation is needed. But there is no experiment involved, the validation part is missing.

Reviewer 4 Report

The paper is very interesting. Application of AI to the material science is now of the great interest of all the world. In my opinion the paper needs a minor revision in the following aspects:

1. The conclusions should include a quantitative analysis of the results. Please add info for example about the accuracy of the results. Also quantitative results should be shown in the abstract.

2. How the validation of the model and verification of the results were carried out. Please add info about it and the reliability of the results.

3. What is the practical application of the method and results presented in the paper? Please add info in Conclusions.

4. Please change the grammar form of the text. In the scientific paper the form "we did" shouldn't be used. Plese use a reported speech in the whole tekst.

Round 2

Reviewer 1 Report

The manuscript reads well and seems better than the previous version.

Reviewer 3 Report

Comments are addressed properly.